# Effect of Hogweed Pectin on Rheological, Mechanical, and Sensory Properties of Apple Pectin Hydrogel

**DOI:** 10.3390/gels9030225

**Published:** 2023-03-14

**Authors:** Sergey Popov, Vasily Smirnov, Daria Khramova, Nikita Paderin, Elizaveta Chistiakova, Dmitry Ptashkin, Fedor Vityazev

**Affiliations:** Institute of Physiology of Federal Research Centre “Komi Science Centre of the Urals Branch of the Russian Academy of Sciences″, 50 Pervomaiskaya Str., 167982 Syktyvkar, Russia

**Keywords:** low-methyl-esterified pectin, hydrogel, ionotropic gelation, texture, rheology, electromyography, sensory analysis, simulated digestion

## Abstract

This study aims to develop hydrogels from apple pectin (AP) and hogweed pectin (HP) in multiple ratios (4:0; 3:1; 2:2; 1:3; and 0:4) using ionotropic gelling with calcium gluconate. Rheological and textural analyses, electromyography, a sensory analysis, and the digestibility of the hydrogels were determined. Increasing the HP content in the mixed hydrogel increased its strength. The Young’s modulus and tangent after flow point values were higher for mixed hydrogels than for pure AP and HP hydrogels, suggesting a synergistic effect. The HP hydrogel increased the chewing duration, number of chews, and masticatory muscle activity. Pectin hydrogels received the same likeness scores and differed only in regard to perceived hardness and brittleness. The galacturonic acid was found predominantly in the incubation medium after the digestion of the pure AP hydrogel in simulated intestinal (SIF) and colonic (SCF) fluids. Galacturonic acid was slightly released from HP-containing hydrogels during chewing and treatment with simulated gastric fluid (SGF) and SIF, as well as in significant amounts during SCF treatment. Thus, new food hydrogels with new rheological, textural, and sensory properties can be obtained from a mixture of two low-methyl-esterified pectins (LMPs) with different structures.

## 1. Introduction

Hydrogels are of increasing interest in food applications, as they make it possible to obtain food that meets modern consumer requirements. Currently, there is an intensive development of new food gels for the creation of healthy food products with high sensory scores. Customizable hydrogel structure design can improve the nutritional value, health content, taste, and texture properties of food-grade hydrogels [1]. Plant-derived polysaccharides have a high potential for the creation of food hydrogels due to their safety and inexpensive production. [2]. Pectin is an acidic polysaccharide that is a component that is located in the spaces between walls of cells of flowering plants. [3]. Pectin-based gelled foods are becoming increasingly popular thanks to the gelling properties and biological activities of pectin [4,5,6].

Pectin is not broken down in the upper gastrointestinal tract and is fermented by colon bacteria [5]. Therefore, pectin has gained a lot of importance in its use as a food matrix for the targeted release of bioactivities to the colon [7]. Nevertheless, the usage of hydrogels made of commercial pectins as colon delivery systems is restricted due to intense swelling, declining mechanical properties, and quick degradation in the small intestine [4]. Citrus and apple pectins are the most commonly used for hydrogel preparation [7]. Generally, the macromolecule of pectin consists of nonbranched and branched areas. The main structural domain of pectin is homogalacturonan, which is made of 1,4-α-D-galacturonic acid residues. The homogalacturonic part largely determines the gel-forming and emulsifying properties of pectin [8]. Therefore, traditional ways of extracting pectin from plant raw material based on long treatment with acids at high temperatures destroy less stable ramified regions so that commercial pectins often consist primarily of linear homogalacturonan [9]. However, many natural pectins additionally contain such structural domains as rhamnogalacturonan II, xylogalacturonan, and others. Moreover, an important branched domain of pectins is rhamnogalacturonan type I in which arabinose and galactose are attached to rhamnose residues embedded in galacturonan, forming side chains [10]. Carboxyls of galacturonic acids are partially methyl-esterified, separating pectins into high-methyl-esterified pectins (HMPs) and LMPs. HMPs form gels at low pH, and gelling LMPs are provided by multivalent cations [11]. Gelling of LMPs involves free carboxyls interacting with cations to create “egg-box″ formations, polysaccharide chain dimerization, and the aggregation of dimers into a gel network [11,12].

Rhamnogalacturonan-I-containing pectin from hogweed H. sosnówskyi (HP) was earlier isolated and characterized [13]. Hogweed pectin was found to be an LMP and formed a stronger calcium hydrogel than commercial apple pectin (AP) [14]. A hydrogel made from HP was found to be more stable than a hydrogel made from AP in saline with pH 3.0–8.0 [15]. However, the degradation of HP hydrogels during digestion has not been investigated yet. We assume here that HP hydrogels are more stable in the gastrointestinal environment than AP hydrogels and that the addition of HP changes the properties of AP hydrogels. It was previously shown that the addition of an HMP to an LMP produced a stronger mixed gel due to a synergistic effect [16], whereas the gel properties of a mixture of two LMPs have not been determined earlier.

This investigation aims to estimate the influence of HP (degree of methyl esterification ca. 21%) on the rheological and mechanical properties of AP (degree of methyl esterification ca. 43%) hydrogels. Chewing, bolus formation, sensory scores, and the simulated digestibility of pure and mixed hydrogels made from AP and HP are also compared.

## 2. Results and Discussion

### 2.1. Preparation of Pectin Hydrogels

Ionotropic hydrogels were created by immersing a dialysis bag containing a pectin solution (50 mL) in a calcium gluconate solution (200 mL). After 48 h, a solid hydrogel was formed in the dialysis bag in the form of a cylinder with a length of 7–8 cm, a diameter of 2.4–2.8 cm, and a weight of 44–48 g. The cylindrical hydrogel was then cut into puck-shaped samples at a predetermined height for subsequent analyses.

Five types of pectin hydrogels (PG1–PG5) were obtained using AP and HP in different ratios (4:0; 3:1; 2:2; 1:3, and 0:4). The PGs had the same density (1.07–1.13 mg/mm^3^) and water content, measured as the weight loss (WL) during drying (80.7–82.7%), regardless of the pectin composition. Increasing the HP content in the hydrogels slightly increased the pH. The pH values for PG1, PG3, and PG5 were 4.42 ± 0.05, 4.55 ± 0.04, and 4.70 ± 0.08, respectively.

An external gelation method used in functional food applications of LMPs is typically based on heating and cooling calcium solutions [5,16]. To encapsulate thermally unstable bioactive substances, pectin gel beads are usually obtained at room temperature by dropping a pectin solution into a solution of calcium ions [17,18,19]. We received pectin hydrogels at room temperature in order to keep the possibility of loading them with functional ingredients open. At the same time, larger sizes of the obtained hydrogel samples than gel beads of a typical diameter of 1–2 mm were more convenient for studying the textural properties of hydrogel. Calcium gluconate rather than calcium chloride was used to improve consumer acceptance of the hydrogels, as calcium chloride has a strong, bitter taste. Sucrose was shown to promote the formation of pectin gels because its hydroxyl groups could stabilize Ca–pectin crosslinks and form hydrogen bonds with water molecules, immobilizing free water and concentrating the polymer environment for gelation [16]. Similar values of hydrogel density and water content among the obtained PGs were expected. First, the hydrogels contained an equal amount of pectin (4%), and second, an equal concentration of sucrose (10%) provided a similar water-keeping capacity.

The pH shift caused by the addition of the HP seemed to be associated with a lower methyl esterification of HP than AP. The greater number of free carboxyl groups in the HP provided more sites of interaction with calcium ions, which apparently led to a higher content of calcium ions in the HP hydrogel than in the AP hydrogel. Therefore, there were fewer dissociated carboxyl groups and fewer free hydrogen ions in HP hydrogel homogenates prepared for pH measurement than in AP ones. As a result, the pH of the HP hydrogel was higher since samples made from HP hydrogel represented the calcium pectate of a weaker acid and stronger base than samples made from AP hydrogel.

### 2.2. Rheological Properties of Pectin Hydrogels

The rheological data were analyzed in the linear (LVE) and nonlinear (n-LVE) viscoelastic regions. The storage modulus G′ of all the PG samples was greater than the loss modulus G″ throughout the LVE region. The G′_LVE_ and G″_LVE_ values increased with increasing HP content in the hydrogels, indicating an increase in network rigidity and total structural strength (Table 1). The G′_LVE_ and G″_LVE_ values of PG5 were 1.9 and 2.3 times higher than those of PG1, respectively. The loss factor, tan [δ]_LVE_, represented a material’s physical behavior. The tan [δ]_LVE_ was 0.20–0.27 for PG1–PG5, indicating that pectin hydrogels had a solid-like behavior. The limiting strain (γL) at which G′ sharply diminished with the increase in strain in PG4 and PG5 was lower than that of PG1. The fracture strain (γFR) measured at the fracture point decreased as the hydrogels’ HP content increased. The γFR value of PG5 was 2.5 times lower than that of PG1 (Table 1). The fracture stress (τFP), where G′ = G″, and the complex modulus (G*FP) at τFP did not differ between PG1 and PG5. The slope of the loss tangent in the nonlinear region (tan [δ] AF) described the transition of the gels from the elastic state to the viscous state since it showed the ratios of the viscous and elastic constituents of materials at large amounts of deformation. PG1, PG2, PG4, and PG5 showed similar tan [δ]_AF_ values without any significant differences among them. PG3 showed a higher tan [δ]_AF_ value than PG1, PG2, PG4, and PG5, which indicated the highest spreadability of the hydrogel containing equal amounts of AP and HP (Table 1).

The rheological parameters of the frequency sweep tests in the LVE range are shown in Table 2. G′ and G″ frequency dependencies are described by the power law equation with high correlation (*R*^2^ = 0.91–0.99). The G′ values increased as the hydrogels’ HP content increased. The G′ values of PG5 were 1.4 times higher than those of PG1. The frequency dependences of the elastic (k′), loss (k″), and complex (A) moduli increased with increase in the HP content in the hydrogels. However, the overall loss tangent (k″/k′) and complex viscosity slope (η*S) did not differ among PG1–PG5. All PGs seemed to show elastic behavior, as revealed by the overall loss tangent (k″/k′) values in the range 0.13–0.19, as well as the low slope of both moduli indicated by 0.08 ≤ *n*′ ≤ 0.10 and 0.04 ≤ *n*″ ≤ 0.10 (Table 2).

The viscosity values of the PGs are given in Table 3.

Using an agglomerative hierarchical cluster analysis, Alghooneh et al. [20] divided the rheological properties of hydrocolloids into properties related to the strength, number, and distance of linkages, as well as the timescale of the junction zone. The rheological parameters of the first group, such as G′_LVE_, k′, k″, and A, increased with increasing HP content in the hydrogels, indicating an increase in the strength of the linkage in the HP-containing hydrogels. The rheological parameters of the group “number of linkages″ (G*max/G*_LVE_, τFr, *n*′, and z), “timescale of junction zone″ (Tan [δ]_LVE_), k″/k′, and η*s), and “distance of linkage″ (*n*″) did not differ among PG1–PG5, indicating the same number of linkages, the same time required for the transformation of hydrogel network chains to the thermodynamically ideal state, and the same average linear distance between two adjacent crosslinks in the gel networks.

### 2.3. Mechanical Properties of Pectin Hydrogels

The PGs showed force–distance curves of very different sizes in the puncture test (Figure 1). Table 4 shows the results of the puncture test of the PGs.

Hardness, measured as the maximum peak force [21], increased with increasing HP content in the hydrogels, indicating better stability of the HP gel networks under large deformation. The hardness value of PG5 was 2.4 times higher than that of PG1. The curve of PG1 did not have a fracture until the maximum peak was reached, resulting in the same hardness and fracturability values for PG1. Fracturability increased as the hydrogels’ HP content increased. The curves of PG3–PG5 exhibited a significant break in the curve before reaching the maximum peak (shown by arrows in Figure 1). The fracturability value of PG5 was 2.3 times higher than that of PG1. The consistency represented the required work for deforming the material [22]. Similar to hardness and fracturability, PG5 and PG1 demonstrated the highest and lowest consistencies, respectively. Further, PG5 showed the highest values of adhesiveness, determined as the maximum negative peak below the baseline during withdrawal. A synergistic effect of AP and HP on the Young’s modulus of mixed hydrogels was discovered. The Young’s modulus [23] was the highest for PG3, which contained equal amounts of AP and HP. The Young’s modulus value of PG3 exceeded those of PG1 and PG5 by 29 and 65%, respectively. Interestingly, the Young’s modulus was positively correlated (*R* = 0.88, *p* < 0.05) with tan [δ]AF.

PG1–PG3 had a close serum release in the range of 0.62–0.71%. The serum release values of PG4 and PG5 were 0.48 ± 0.08 and 0.49 ± 0.03% (*p* < 0.05 vs. PG1–PG3), respectively.

The changes in mechanical parameters (hardness and consistency) were consistent with the changes in rheological parameters and confirmed a stronger linkage in HP hydrogels compared to AP hydrogels [24]. There are conflicting data for the effect of textural hardness on liking and acceptability evaluations of hydrogel foods. In a number of studies, increases in the hardness of hydrogels have led to decreases in their liking scores [25,26,27,28]. However, studies [29,30] have reported an increase in overall liking of foods with higher hardness, whereas others [31,32,33] have found no effect of hardness on acceptability. Increasing food hardness may be effective at slowing energy intake [34], and therefore, HP hydrogels may be useful for enhancing satiety and combating obesity.

Hydrogels containing HP demonstrated higher fracturability and adhesiveness than AP hydrogels, similar to hardness and consistency. Fractures can occur when all the bonds in a gel network are broken [20], so the increase in fracturability indicated higher numbers of linkages in the HP hydrogels. It should be noted, however, that G*max/G*LVE, τFr, *n*′, and z values, which were related to the number of linkages, did not differ among PG1–PG5. Increased adhesiveness indicated an increase in the linkage distance, which contradicted the rheological analysis, which revealed equal *n*″ values in all the PGs. Different degrees of deformation in mechanical and rheological tests could explain the discrepancy in results.

Differences in the chemical structures of HP and AP apparently determined the different properties of pectin hydrogels PG1–PG5, which were made of different contents of HP and AP. It was assumed that Ca^2+^ ions bound to free COO groups in the homogalacturonic region. Then, dimerization of polysaccharide chains and dimer–dimer interactions occurred due to hydrogen bonds [11,12]. The lower degree of methyl esterification seemed to provide a higher density of ionic crosslinks in the HP compared to the AP hydrogel network (Figure 2). In addition, HP macromolecules had a higher content of arabinose residues (3.3 vs. 0.8 mol%), which indicated a more pronounced rhamnogalacturonan I region than that of AP. The arabinose-modulated gelling of pectin because of hydrogen bonding interactions provided additional entanglements [35,36]. In a study [35], it was shown that a polysaccharide with a halved content of arabinose yielded a less strong gel. Zheng et al. [36] reported increased gel strength because of the limitation of network chain mobility provided by arabinose residues. Therefore, pectin hydrogel enhancement by HP may also be due to arabinose of the side chains, which added to the effect of more free carboxyl groups in the backbone (Figure 2).

### 2.4. Oral Processing and Sensory Evaluation of Pectin Hydrogels

Electromyography (EMG) parameters did not differ during the chewing of PG1 and PG3 (Figure 3). Chewing PG5 required 19% more time (Figure 3A) and 20% more chews (Figure 3B) than PG1 and PG3. Masseter and temporalis activities were 30% higher when chewing PG5 than when chewing PG1 and PG3 (Figure 3C).

As known, the incorporation of saliva into a bolus while chewing transforms the initial value of moisture content in solid or semisolid foods to prepare a ready-to-swallow bolus in the mouth [37]. A gravimetrical analysis of saliva content in the bolus formed from the PGs revealed that saliva uptake varied between 22.6 and 29.1 wt% in pectin hydrogels (Table 5).

Saliva uptake has been found to vary between 3 and 10 wt% in different gels [38,39]. PG3 failed to affect salivation compared to PG1 (Table 5). A 29% increase in saliva in the PG5 bolus was comparable to that of PG1. Because the compositions of the PGs were similar, serum release appeared to be a critical initial characteristic of pectin hydrogels that could affect saliva secretion during oral processing and subsequent saliva uptake into the bolus. Saliva not only hydrates the bolus but also lubricates it by lowering friction due to its watery character and the presence of salivary mucins, thereby making swallowing the bolus safe [37,40]. Therefore, PG5, with its low serum release (0.49%), needed more saliva incorporation to enhance lubrication than PG1 (serum release 0.66%). Similar to our findings, low serum release agar-gelatin gels incorporated more saliva into the bolus [38]. The highest increase in saliva uptake was observed for the bolus of the hardest PG5 samples. It might seem intuitively obvious that the textural differences between PG1 and PG5 determined salivary flow while chewing hydrogels. However, the saliva incorporation rate (SIR) into the bolus was similar for all the PGs (Table 5). Obviously, the hardest PG5 samples needed more chews to swallow, likely due to the more complex microstructure of the PG5 mesh, which required more mechanical mixing and saliva incorporation to form a cohesive bolus. This result is consistent with a previous study on carrageenan gels [41], which reported that prolonged chewing time increased saliva incorporation. The boluses demonstrated shear-thinning behavior (Figure 4), which is in agreement with previous results [42].

A resulting power law index η and consistency parameter K for the bolus are shown in Table 5. All volunteers presented a safe and efficacious swallow of the bolus from all the PGs. The viscosity of the bolus dramatically decreased under mastication and was significantly lower than the initial viscosity of the pectin hydrogel samples at orally relevant shear rates (50 s^−1^) (Table 3), although the viscosity values were higher than those of thickened fluids for dysphagia management [43]. The viscosity values were not significantly different among three PGs at the swallowing point, despite the differences in the initial viscosity, when volunteers were given PG1, PG3, and PG5 with viscosities of 273.9, 291.4, and 502.0 Pa*s at shear rates of 50 s^−1^, respectively (Table 3). In practice, subjects adjust the duration and intensity of chewing to obtain a bolus that is easy to swallow. [39,44,45]. Since samples of PG5 were the hardest, volunteers adapted their chewing to obtain boluses with similar rheological characteristics, regardless of hydrogel composition. The increase in saliva uptake in the PG5 bolus indicates that more saliva was incorporated into the bolus to give it the viscosity that was needed for safe and efficient swallowing. Thus, a long duration of chewing, more chewing movements, and higher muscle activity while chewing PG5 appeared to be required to achieve comfortable swallowing characteristics of the bolus.

Sixteen nontrained subjects rated the overall and consistency liking using a nine-point hedonic scale and the perceived intensity of texture attributes using a 100 mm visual analog scale for the sensory evaluation of pectin hydrogels. PG1, PG3, and PG5 received the same likeness scores, which lay between slightly disliking (score ≃ 4) and neither liking nor disliking (score ≃5). Figure 5 shows the mean values of the perceived texture scores during the early (hardness, brittleness, springiness, and moisture), middle (adhesiveness and chewiness), and late (easiness to swallow) chewing of the PGs.

The PGs differed only in regard to perceived hardness and brittleness. The perceived hardness of PG1 and PG5 did not differ, while the perceived hardness of PG3 was lower by 35 and 48% than that of PG1 and PG5, respectively. The perceived brittleness of PG1 and PG3 was similar, whereas the perceived brittleness of PG5 was lower by 33 and 29% than that of PG1 and PG3, respectively. An unexpected result was that, in our study, the sensory assessments of hardness and brittleness did not coincide with the data from the instrumental analysis.

### 2.5. Correlation Analysis of Mechanical, Rheological, and Sensory Properties of Pectin Hydrogels

The interrelationships between the sensory attributes and the mechanical and rheological properties of PGs were investigated using Pearson’s coefficient (Table 6). Perceived and instrumental hardness were not correlated, while perceived and instrumental brittleness were inversely correlated. Consistent with [42,46], perceived hardness was positively correlated with the viscosity of the pectin hydrogels. A positive relationship among the perceived hardness, elasticity, and adhesiveness of the PGs was found. Young’s modulus and consistency were inversely related to perceived hardness. Fracturability was inversely correlated with almost all the rheological parameters, while chewiness was positively correlated with almost all the rheological parameters, which is consistent with a previous study [42].

Correlations among the liking and sensory-textural attributes are presented in Table 7. Overall liking was shown to positively correlate with consistency liking, taste, and ease to swallow. In turn, liking consistency was positively related to perceived brittleness, moisture, and swallowing ease. In agreement with the EMG data, perceived hardness and chewiness were positively correlated. An inverse correlation among perceived hardness, moisture, and ease of swallowing was revealed. Perceived brittleness was positively correlated with moisture (Table 7).

Several studies have previously demonstrated the interrelationships between the liking and sensory textural attributes of food gels. A study [47] discovered a positive relationship between the hardness and chewability of k-carrageenan hydrogels and k-carrageenan mixed with sodium alginate hydrogels. A sensory analysis of gels made from three hydrocolloids showed a positive correlation between firmness and elasticity, chewiness, and cohesiveness [42].

The correlation analysis of sensory attributes and oral-processing parameters revealed a positive relationship between perceived springiness and temporal muscle activity, as well as between perceived adhesiveness and bolus viscosity (Table 8). Chewiness was positively correlated with masseter muscle activity and saliva uptake, but it was inversely correlated with bolus viscosity. Easy swallowing was positively correlated with masseter and temporal muscle activity.

### 2.6. Simulated Digestibility of Pectin Hydrogels

The content of total sugars gradually decreased during the digestion of PG1, PG3, and PG5 (Figure 6).

Galacturonic acid was released predominantly after the digestion of PG1 in SIF and SCF (Figure 7A). Galacturonic acid was slightly released from PG5 during the initial stages of digestion (OP, SGF, and SIF) and in significant amounts during the SCF phase. The release of galacturonic acid from PG3 had an intermediate intensity compared to PG1 and PG5 in SIF and did not increase in SCF compared to SIF. During digestion in OP and SGF, calcium was intensively released from the PGs regardless of pectin composition, probably due to the strong destruction of the hydrogels’ structure during chewing in vivo (Figure 7B). During digestion in SIF, calcium was released from PG3 and PG5 6.3 and 4.8 times less than from PG1, respectively.

The release of calcium during in vivo chewing could be attributed to the release of free calcium contained in the gel matrix rather than crosslinked calcium. The concentration of crosslinked calcium for PG preparation was excessive and corresponded to a stoichiometric ratio (R = 2(Ca^2+^)/(COO^−^)) in the range of 4.1–5.7. The strongest gel was formed at R = 2–3 [48]. It was also suggested that high calcium concentrations may hinder the formation of gel networks when R >> 1 [48].

Similarly, the large amounts of total sugars in the first phases of digestion were likely due to the release of sucrose and not to the destruction of gel network polysaccharides. According to galacturonic acid release data, the HP gel network was more resistant to digestion than the AP gel network. The more stable HP hydrogel in the upper gastrointestinal tract appeared to be a better fit than the AP hydrogel as a vehicle for the delivery of biologically active substances to the colon.

## 3. Conclusions

The prospects for the use of HP as a hydrocolloid for the production of food gels were evaluated. The advantages of HP food hydrogels compared with AP food hydrogels were demonstrated regarding stronger structure and higher digestion resistance. Rheological and textural measurements showed that the strength and number of linkages in the HP hydrogels were higher than in the AP hydrogels, so increasing the HP content in the mixed hydrogel increased its strength. The synergistic gelation between pectins occurred because the Young’s modulus and tangent after flow point values were higher for mixed hydrogels that contained equal amounts of AP and HP than for pure AP and HP hydrogels. Stronger HP hydrogels increased chewing duration and intensity. More intense and longer oral processing of HP hydrogels provided a ready-to-swallow bolus with the same viscosity and salivary wetness as an AP hydrogel bolus. Untrained volunteers gave pectin hydrogels high marks for overall and texture liking, indicating that they had a high potential for consumer acceptance. Among the sensory textural attributes, HP was characterized by reduced brittleness, which had a an inverse correlation with almost all the rheological and mechanical properties. The main functional advantage of HP-containing hydrogels was their resistance to digestion in the small intestine. Therefore, HP hydrogels could be promising as a food matrix carrier of biologically active substances delivered to the large intestine. Thus, new food hydrogels with new rheological, textural, and sensory properties could be obtained from a mixture of low-methyl-esterified pectins with different structures.

## 4. Materials and Methods

### 4.1. Materials

The isolation and chemical properties of HP were previously described [14]. Apple pectin AU701 (Herbstreith & Fox GmbH, Neuemberg, Germany) was used as AP. Table 9 shows the characteristics of AP and HP. Calcium gluconate was supplied by Zhejiang Tianyi Food Additives (Tongxiang, Zhejiang, China); sucrose was purchased from a local supermarket.

### 4.2. Preparation of Pectin Hydrogels

AP, HP (2 g), three different blend combinations of AP (1.5 g) and HP (0.5 g), AP (1 g) and HP (1 g), or AP (0.5 g) and HP (1.5 g) were dissolved in deionized water (50 mL) with the addition of sucrose (5 g). The solutions were heated (60 °C) under continuous magnetic stirring (200 rpm) for 60 min for better dissolution and then cooled to room temperature. Dialysis tubes with a pore size of 14 kPa (Sigma-Aldrich Co, St. Louis, UO, USA) were filled with the solutions obtained (50 mL) and were held in a solution of 0.3 M calcium gluconate and 10% sucrose (200 mL) for 48 h at 25 °C for pectin gelling (Figure 8A).

Cylinder-shaped (7–8 cm high) hydrogels were formed (Figure 8B). After being removed from the dialysis tubes, the hydrogels were washed with distilled water and cut into single-serving puck-shaped pieces at a predetermined height for subsequent analyses (Figure 8C). Five types of pectin hydrogels were obtained depending on the content of AP and HP in the pectin mixture: PG1—4% AP; PG2—3% AP mixed with 1% HP; PG3—2% AP mixed with 2% HP; PG4—1% AP mixed with 3% HP; and PG5—4% HP (Table 10).

### 4.3. Characterization of Pectin Hydrogels

#### 4.3.1. General Characterization

The pH was determined for hydrogel aqueous homogenates (1:10 (*w*/*v*)) using an S20 SevenEasy™ pH meter (Mettler-Toledo AG, Schwerzenbach, Switzerland). The weight of 1 cm square hydrogel cubes (*n* = 8) was measured (AG245, Mettler Toledo International, Greifensee, Switzerland) to determine the density as weight/volume. Weight loss (WL) was determined using a gravimetric method [15].

#### 4.3.2. Rheological Characterization of Pectin Hydrogels

A rotational-type rheometer (Anton Paar, Physica MCR 302, Graz, Austria) equipped with a parallel plate geometry (diameter 25 mm; gap 4.0 mm) was used for the strain and frequency sweep measurements.

Strain sweep evaluation was performed from 0.01 to 100% strain amplitudes using a controlled shear rate mode at 20 °C at a constant frequency and stress of 1 Hz and 9.0 Pa, respectively. The storage modulus (G′_LVE_), loss modulus (G″_LVE_), loss tangent (tan δ) in LVE, complex modulus (G*_LVE_), limiting value of strain (the strain at which biopolymers enter from linear viscoelastic region to nonlinear viscoelastic region, γL), and stress at flow point (τFP) with the corresponding complex modulus (G*_FP_), fracture stress (τFr), and slope of the loss tangent after flow point (tan δ_AF_) were determined [20]. The shear strain dependence of G′ and G′ was determined using the power law equation:G′ = A′ × 𝜔^n′^,(1)
G″ = A″ × 𝜔^n″^,(2)
where ω is the angular shear strain (%); A′ (Pa*s) and A″ (Pa*s) are intercepts; and n′ and *n*″ are the slopes of G′ and G″ frequency dependence, respectively. A′ = A″ is a measure of the contribution of the viscous component in relation to the elastic component and represents the overall loss tangent of the material.

For the frequency sweep experiments, the obtained mechanical spectra were characterized by the values of G′ and G″ (Pa) as a function of frequency in the range of 0.3–70.0 Hz at 20 °C and a constant stress of 9.0 Pa. The loss factor tan δ was calculated as the ratio of G″ and G′ [49]. The power law function [50] was expressed as follows:η = K_c_ × *y*^n^,(3)
where η is the steady viscosity, K_c_ is the consistency constant, *y* is the shear rate, and *n* is the power law index or flow behavior index.

The degrees of frequency dependence for G′ and G″ were determined by the power law parameters [51], which are expressed as follows:G′ = k′ × 𝜔^n′^,(4)
G″ = k″ × 𝜔^n″^,(5)
where G′ and G″ are the storage and loss moduli, respectively; 𝜔 is the oscillation frequency (Hz); and k′ and k″ are constants. In addition, the complex dynamic viscosity frequency dependence η*s was determined.

The strength of the network (A, Pa*s^1/z^) and the network extension parameter (z) were evaluated as follows according to [52]:G′ = G* × (ω) = A × ω^1/z^,(6)
where G* (Pa) is the complex modulus.

#### 4.3.3. Instrumental Texture Characterization of Pectin Hydrogels

A puncture test (probe P/5 mm, depth 4 mm) for the PG1, PG2, PG3, PG4, and PG5 samples (0.4 cm high) was carried out using a TA-XT Plus (Texture Technologies Corp., Stable Micro Systems, Godalming, UK) instrument at room temperature.

Serum release was determined as the weight ratio of the released serum to the initial weight of a hydrogel after compression to 80% of its original height at room temperature [53].

#### 4.3.4. Characterization of Oral Processing of Pectin Hydrogels

Sixteen volunteers of both sexes without masticatory or swallowing dysfunctions were involved equally in the research. Three test sessions were performed by each subject: (1) EMG recording under unilateral chewing; (2) sensory score assessment; and (3) bolus collection after free chewing. Puck-shaped pieces (~6 g, 1 cm high) of the hydrogel samples (PG1, PG3, and PG5) were presented to each participant on a plastic spoon.

EMG activity from the superficial masseter and anterior temporalis was monitored using bipolar electrodes (11 × 25 mm) that were separated from each other by approximately 20 mm. Before EMG recording, participants were instructed to chew freely until it was easy to swallow [53].

Then, participants evaluated nine sensory attributes using a 100 mm visual analog scale and acceptability using a 9-point hedonic scale [54].

In the third session, participants were asked to chew a fixed, preweighed quantity (6 g, 1 cm high) of PG1, PG3, or PG5 samples in a single mouthful and spit out the bolus just before swallowing it. The weight of the wet bolus was calculated as the combined weight of the expectorated material minus the weight of rinsing water [55]. Salivary uptake was calculated using the following equation:SU = (WB − WH)/WH × 100,(7)
where SU is saliva uptake (%), and the WB and WH are weights of the wet bolus (g) and wet hydrogel sample (g), respectively. The rate of saliva incorporation (SIR) was calculated using the following equation:SIR = (WB − WH)/TC,(8)
where SIR is saliva incorporation (g/min), TC is the time of chewing (min), and WB and WH are weights of the wet bolus (g) and wet hydrogel sample (g), respectively.

The apparent viscosities (flow curves) of the bolus were obtained at shear rates ranging from 0.0001 to 100 s^−1^ at 37 °C using a rotational-type rheometer [42].

#### 4.3.5. In Vivo Oral Phase (OP) and Static In Vitro Gastrointestinal Digestion

Six healthy volunteers chewed each type of hydrogel (4 g) 20 times and spat the bolus into a beaker [56]. Immediately thereafter, 4.0 mL of water was added to the beaker, it was stirred, and all the fluid was separated for analysis. The gel fragments were transferred to a 20 mL sheathed glass container for further in vitro digestion. In vitro gastrointestinal digestion was approached by a method [53] using SGF (pH 1.5, 0.08 M HCl, and 0.03 M NaCl), SIF (pH 6.8, 0.05 M KH_2_PO_4_, and 0.02 M NaOH), and SCF (0.01 M KH_2_PO_4_, 0.05 M NaHPO_4_, and pectinase: 1.7 mg/mL). The contents of galacturonic acid and calcium ions were determined in the fluid after each phase of digestion. For this, aliquots (1–2 mL) of incubation medium were taken and centrifuged, and the resulting supernatant was precipitated with a fourfold volume of 96% ethanol. The precipitate was washed twice with 96% ethanol and dissolved with 3 mL of H_2_O. The resulting solution was used to determine the content of galacturonic acid by the reaction of the sample with 3,5-dimethylphenol in the presence of concentrated H_2_SO_4_ [57]. The alcohol supernatant was used to determine the total amount of sugars using the phenol-sulfur method. The concentration of calcium was determined using a Calcium-Agat kit (Agat-Med, Moscow, Russia).

### 4.4. Statistical Analysis

All statistical analyses were performed using Statistica 10.0 (StatSoft, Inc., Tulsa, OK, USA). Results are presented as means ± standard deviations (SDs). The differences among the means in serum release, the rheological and mechanical parameters, and the digestion studies were estimated with one-way ANOVA and Tukey’s HSD test. A one-way repeated ANOVA and Fisher’s LSD post hoc test were applied to determine differences in the EMG, sensory, saliva, and bolus variables for different hydrogels. Pearson’s correlations were calculated to study the relationships among the rheological, mechanical, and sensory properties of the hydrogels.

## Figures and Tables

**Figure 1 gels-09-00225-f001:**
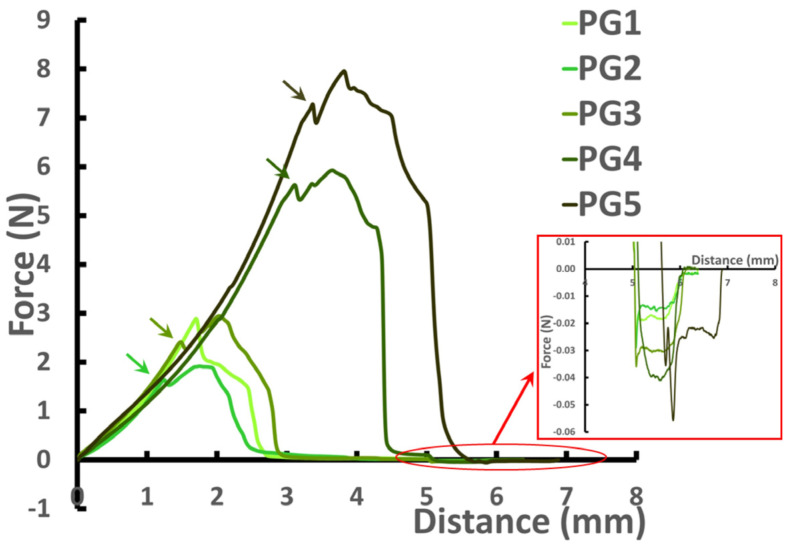
Representative force–distance curves of pectin hydrogels PG1–PG5. The arrows show the first significant break. The insertion window demonstrates the curve area, which corresponded to adhesiveness.

**Figure 2 gels-09-00225-f002:**
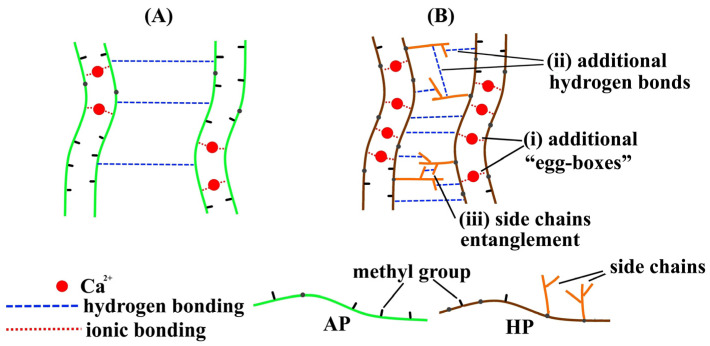
Scheme of a proposed chain interaction in AP (**A**) and HP (**B**) hydrogels. Suggested reasons for strengthening the HP hydrogel: (i) an increase in the number of Ca^2+^ crosslinks due to a lower degree of esterification; (ii) the formation of additional hydrogen bonds due to more sugar side chains; and (iii) the entanglement of sugar side chains.

**Figure 3 gels-09-00225-f003:**
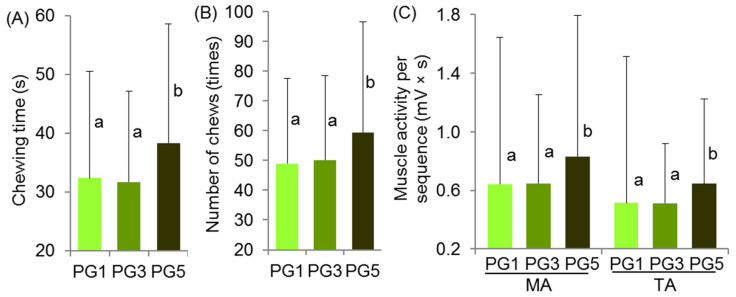
Electromyography data on PG chewing time (A), number (B), and activity of the masseter (MA) and temporal (TA) muscles (C). Means with standard deviations are given. Differences are significant (*p* < 0.05) between means labeled with different lowercase letters (a and b) (*n* = 16).

**Figure 4 gels-09-00225-f004:**
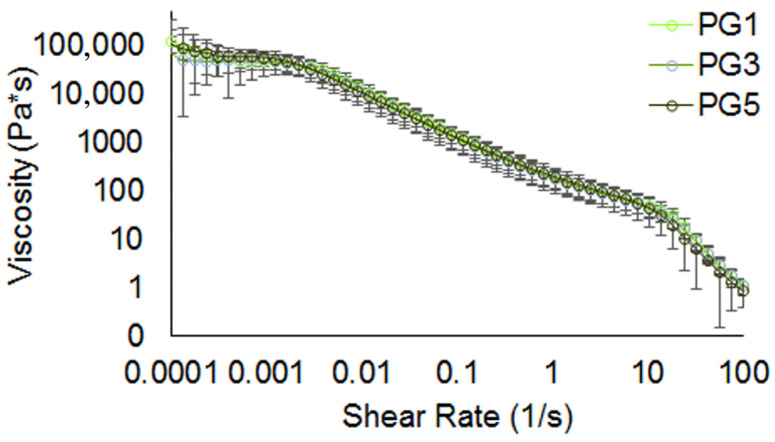
Flow curves of boluses from PGs.

**Figure 5 gels-09-00225-f005:**
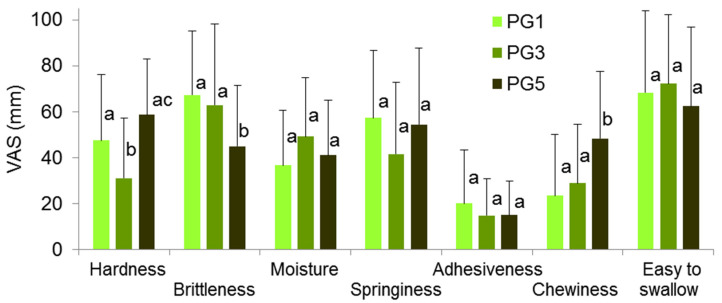
Sensory attributes of PGs. Means with standard deviations are given. Differences are significant (*p* < 0.05) between means labeled with different lowercase letters (a, b, and c) (*n* = 16).

**Figure 6 gels-09-00225-f006:**
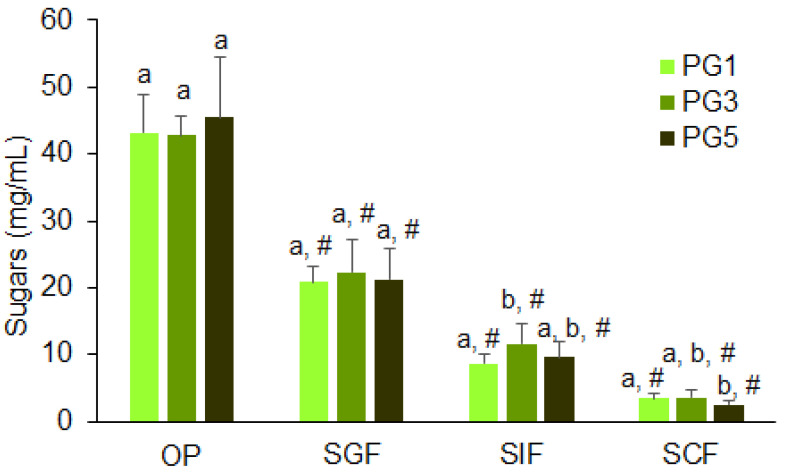
The amount of total sugars measured using phenol-sulfur method released from PGs during successive oral in vivo (OP) and gastrointestinal in vitro phases of digestion. SGF, SIF, and SCF—simulated gastric, intestinal, and colonic fluids, respectively. Means with standard deviations are given. Differences are significant (*p* < 0.05) between means labeled with different lowercase letters (a and b); #—indicates significant differences vs. the previous phase (*n* = 6, *p* < 0.05).

**Figure 7 gels-09-00225-f007:**
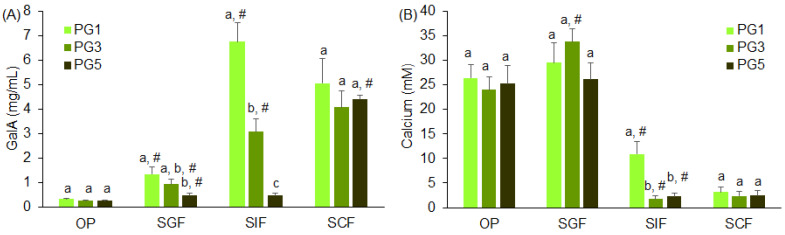
The amounts of galacturonic acid (GalA) (**A**) and calcium (**B**) released from PGs during successive oral in vivo (OP) and gastrointestinal in vitro phases of digestion. SGF, SIF, and SCF—simulated gastric, intestinal, and colonic fluids, respectively. Means with standard deviations are given. Differences are significant (*p* < 0.05) between means labeled with different lowercase letters (a, b, and c); #—indicates significant differences vs. the previous phase (*n* = 6, *p* < 0.05).

**Figure 8 gels-09-00225-f008:**
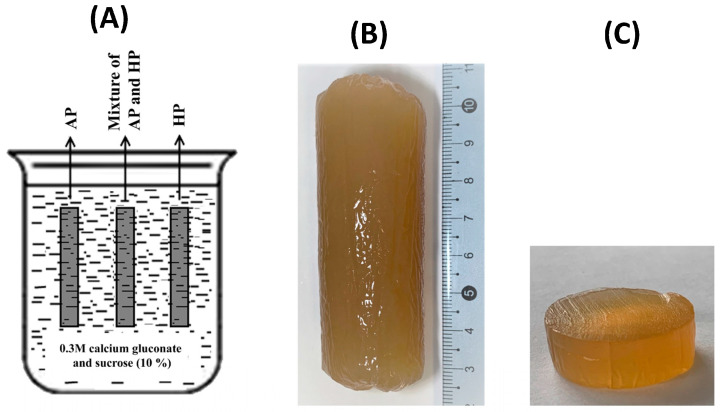
Preparation of hydrogels. General scheme (**A**), a representative image of PG3 hydrogel formed in a dialysis bag (**B**), and puck-shaped PG3 sample used for analyses (**C**).

**Table 1 gels-09-00225-t001:** Strain sweep for pectin hydrogels PG1–PG5 in the amplitude sweep test (1 Hz. 20 °C).

Parameters	PG1	PG2	PG3	PG4	PG5
G′_LVE_ (Pa)	51,636 ± 4741 ^a^	61,209 ± 7204 ^b^	74,486 ± 10,993 ^c^	88,753 ± 5467 ^d^	98,533 ± 14,595 ^e^
G″_LVE_ (Pa)	10,966 ± 2033 ^a^	12,997 ± 3039 ^a^	16,562 ± 4640 ^b^	17,428 ± 2411 ^c^	25,357 ± 4587 ^d^
G*_LVE_	53,643 ± 5536 ^a^	63,046 ± 5936 ^b^	77,100 ± 9036 ^c^	89,529 ± 6592 ^d^	101,313 ± 15,488 ^e^
Tan [δ]_LVE_	0.21 ± 0.04 ^a^	0.22 ± 0.09 ^a^	0.24 ± 0.12 ^a^	0.20 ± 0.04 ^a^	0.27 ± 0.10 ^a^
γL (%)	0.35 ± 0.08 ^a^	0.23 ± 0.02 ^a,b^	0.20 ± 0.01 ^a,b^	0.19 ± 0.02 ^b^	0.15 ± 0.04 ^b^
τFr (Pa)	251 ± 151 ^a^	155 ± 48 ^a^	170 ± 18 ^a^	216 ± 42 ^a^	114 ± 24 ^a^
γFr	0.75 ± 0.10 ^a^	0.54 ± 0.03 ^b^	0.48 ± 0.05 ^b,c^	0.44 ± 0.01 ^c^	0.30 ± 0.06 ^d^
G*_FP_ (Pa)	24,252 ± 9158 ^a^	29,605 ± 8375 ^a^	38,404 ± 5752 ^a^	47,546 ± 9452 ^a^	53,702 ± 39,039 ^a^
Tan [δ]_AF_	0.34 ± 0.13 ^a,c,d^	0.36 ± 0.12 ^a,b,c,d^	0.66 ± 0.06 ^b^	0.39 ± 0.06 ^a,c^	0.19 ± 0.04 ^a,d^
G*max/G*_LVE_	1.25 ± 0.21 ^a^	1.26 ± 0.17 ^a^	1.20 ± 0.02 ^a^	1.05 ± 0.08 ^a^	1.53 ± 0.18 ^a^

Means with standard deviations are given (*n* = 3). Differences are significant (*p* < 0.05) between means labeled with different lowercase letters (a, b, c, and d). Storage modulus (G′_LVE_), loss modulus (G″_LVE_), complex modulus (G*_LVE_), limiting value of strain (γL), loss tangent (TAN [δ]_LVE_), fracture stress (τFr), corresponding complex modulus (G*_FP_) with the stress at flow point, and slope of the loss tangent after flow point (tan [δ]AF).

**Table 2 gels-09-00225-t002:** Frequency dependence of viscoelastic parameters for pectin hydrogels PG1–PG5, viscosity, and frequency (0.3 < *ω* < 70.0 or at 0.54/1.11/10.50/30.60 Hz).

Parameters			PG1	PG2	PG3	PG4	PG5
k′ (Pa*s)			71,741 ± 20,277 ^a^	86,851 ± 9022 ^a^	88,590 ± 3574 ^a^	83,490 ± 32,656 ^a,b^	127,855 ± 15,415 ^b^
k″ (Pa*s)			10,307 ± 2719 ^a^	11,295 ± 2339 ^a^	13,227 ± 92 ^a,b^	15,140 ± 4187 ^a,b^	19,114 ± 5748 ^b^
A			75,313 ± 16,622 ^a^	83,298 ± 5382 ^a^	84,316 ± 4866	101,487 ± 37,040 ^a,b^	135,162 ± 25,722 ^b^
k″/k′			0.14 ± 0.00 ^a^	0.13 ± 0.02 ^a^	0.15 ± 0.01 ^a,b^	0.19 ± 0.03 ^b^	0.17 ± 0.02 ^a,b^
η*_s_			12,039 ± 2919 ^a^	12,701 ± 2685 ^a^	10,096 ± 4678 ^a^	12,346 ± 4791 ^a^	18,773 ± 6072 ^a^
*n*′			0.08 ± 0.02 ^a^	0.09 ± 0.02 ^a^	0.10 ± 0.02 ^a^	0.08 ± 0.05 ^a^	0.09 ± 0.01 ^a^
*n*″			0.10 ± 0.02 ^a,c^	0.10 ± 0.01 ^a^	0.08 ± 0.01 ^a,c^	0.04 ± 0.01 ^b^	0.08 ± 0.05 ^a,b,c^
z			14.2 ± 3.7 ^a^	10.8 ± 3.0 ^a^	9.6 ± 2.4 ^a^	9.3 ± 1.2 ^a^	9.7 ± 1.3 ^a^
Frequency (Hz)	0.54	G′ (kPa)	72.8 ± 18.1 ^a^	74.4 ± 15.7 ^a,b^	62.3 ± 29.5 ^ab^	78.7 ± 32.3 ^a,b^	118.9 ± 25.7 ^b^
		G″ (kPa)	11.3 ± 2.4 ^a^	12.9 ± 3.7 ^a,b^	12.9 ± 4.2 ^b^	19.6 ± 2.9 ^b^	31.8 ± 6.3 ^c^
	1.11	G′ (kPa)	72.4 ± 20.7 ^a^	81.0 ± 17.0 ^a^	69.1 ± 28.0 ^a^	89.2 ± 32.7 ^a^	132.5 ± 27.6 ^b^
		G″ (kPa)	10.4 ± 2.9 ^a^	11.1 ± 2.1 ^a,b^	10.7 ± 4.0 ^ab^	13.7 ± 3.7 ^a,b^	19.2 ± 6.2 ^b^
	10.50	G′ (kPa)	87.1 ± 26.7 ^a^	96.9 ± 19.4 ^a^	84.0 ± 34.8 ^a^	100.4 ± 34.8 ^a,b^	174.2 ± 36.6 ^b^
		G″ (kPa)	13.0 ± 4.1 ^a^	14.1 ± 2.9 ^a^	12.1 ± 4.8 ^a^	16.8 ± 5.1 ^a,b^	26.9 ± 5.6 ^b^
	30.60	G′ (kPa)	94.7 ± 31.6 ^a^	108.3 ± 22.7 ^a^	93.0 ± 38.9 ^a^	100.9 ± 37.7 ^a^	192.0 ± 36.4 ^b^
		G″ (kPa)	15.7 ± 5.9 ^a^	16.0 ± 3.5 ^a^	13.7 ± 5.6 ^a^	17.8 ± 5.8 ^a,b^	30.7 ± 8.8 ^b^

Means with standard deviations are given (*n* = 3). Differences are significant (*p* < 0.05) between means labeled with different lowercase letters (a, b, and c). The frequency dependences of the elastic (k′ and *n*′), loss (k″ and *n*″), and complex (A and z) moduli; overall loss tangent (*k*′/k″); and the slope of complex viscosity (η*S).

**Table 3 gels-09-00225-t003:** Summary of power law parameters for the relationship between storage modulus or viscosity and frequency (0.03 < ω < 70.00 Hz or at 10 and 50 Hz) of pectin hydrogels PG1–PG5.

Hydrogel	Viscosity				
	K (Pa*s)	*R* ^2^	*n*	η_app_ 10 (Hz)	η_app_ 50 (Hz)
PG1	11,942	0.999	−0.946	1333.10 ± 409.6	273.9 ± 109.4
PG2	12,677	0.999	−0.904	1482.5 ± 297.0	343.6 ± 79.6
PG3	10,780	0.999	−0.899	1285.1 ± 531.2	291.4 ± 125.1
PG4	13,310	0.997	−0.926	1541.4 ± 532.7	303.2 ± 111.2
PG5	18,847	0.999	−0.908	2162.8 ± 872.7	502.0 ± 198.3

**Table 4 gels-09-00225-t004:** Mechanical properties of pectin hydrogels PG1–PG5 in the puncture test.

Hydrogel	Hardness (N)	Fracturability (N)	Consistency (mJ)	Adhesiveness (mN)	Brittleness (mm)	Young’s Modulus (kPa)
PG1	3.09 ± 0.41 ^a^	3.09 ± 0.41 ^a^	3.90 ± 0.51 ^a^	30.1 ± 3.4 ^a^	1.9 ± 0.2 ^a^	608 ± 52 ^a^
PG2	2.63 ± 0.49 ^a^	2.56 ± 0.57 ^a^	3.50 ± 0.44 ^a^	26.7 ± 5.7 ^a^	1.7 ± 0.1 ^b^	521 ± 103 ^b^
PG3	4.09 ± 0.98 ^b^	3.93 ± 0.97 ^b^	6.23 ± 1.36 ^b^	39.5 ± 5.1 ^b^	2.1 ± 0.3 ^a^	782 ± 91 ^c^
PG4	6.16 ± 0.70 ^c^	6.04 ± 0.71 ^c^	13.49 ± 2.69 ^c^	50.7 ± 16.1 ^b,c^	3.5 ± 0.2 ^c^	488 ± 161 ^a,b^
PG5	7.37 ± 1.82 ^c^	7.03 ± 1.71 ^c^	18.06 ± 5.29 ^d^	52.5 ± 11.4 ^c^	4.1 ± 0.6 ^d^	474 ± 188 ^a,b^

Means with standard deviations are given (*n* = 8). Differences are significant (*p* < 0.05) between means labeled with different lowercase letters (a, b, c, and d).

**Table 5 gels-09-00225-t005:** Salivary parameters and viscosity of the bolus from PG1, PG3, and PG5.

Hydrogel	Saliva Parameters		Viscosity Parameters				
	SU * (%)	SIR ** (g/min)	K (Pa × s)	*R* ^2^	*n*	η_app_ 10 (s^−1^)	η_app_ 50 (s^−1^)
PG1	22.6 ± 14.0 ^a^	3.04 ± 2.6 ^a^	194.49	0.972	−0.788	50.6 ± 22.1 ^a^	2.8 ± 1.1 ^a^
PG3	24.0 ± 12.7 ^a^	3.42 ± 2.7 ^a^	181.39	0.962	−0.778	47.2 ± 19.7 ^a^	2.9 ± 0.9 ^a^
PG5	29.1 ± 13.4 ^b^	3.38 ± 2.2 ^a^	170.65	0.974	−0.811	43.5 ± 25.7 ^a^	2.1 ± 0.7 ^a^

Means with standard deviations are given. * SU—saliva uptake; ** SIR—saliva incorporation rate. Differences are significant (*p* < 0.05) between means labeled with different lowercase letters (a and b) (*n* = 15).

**Table 6 gels-09-00225-t006:** Correlations between volunteers’ evaluations of texture and the mechanical and rheological properties of PGs.

	Sensory Attributes ^1^						
	1	2	3	4	5	6	7
General properties							
SR	−0.26	0.31 *	−0.01	−0.03	0.09	−0.38 *	0.10
WL	−0.29 *	−0.01	0.21	−0.22	−0.11	0.01	0.07
Rheological properties							
G′LVE (Pa)	0.17	−0.30 *	0.07	−0.03	−0.11	0.36 *	−0.07
G″LVE (Pa)	0.21	−0.31 *	0.05	−0.01	−0.11	0.37 *	−0.08
γFR	−0.12	0.28	−0.10	0.06	0.12	−0.34 *	0.06
k′ (Pa*s)	0.21	−0.31 *	0.05	−0.01	−0.11	0.37 *	−0.08
k″ (Pa*s)	0.23	−0.31 *	0.04	0.01	−0.10	0.37 *	−0.09
ƞ*s	0.37 *	−0.29 *	−0.08	0.12	−0.04	0.34 *	−0.12
A	0.29 *	−0.31 *	0.00	0.05	−0.08	0.37 *	−0.10
Viscosity							
ƞ10	0.34 *	−0.31 *	−0.04	0.09	−0.06	0.36 *	−0.11
ƞ55	0.31 *	−0.31 *	−0.02	0.06	−0.07	0.37 *	−0.11
Mechanical properties							
Hardness	0.26	−0.31 *	0.01	0.03	−0.09	0.38 *	−0.10
Fracturability	0.27	−0.31 *	0.01	0.03	−0.09	0.38 *	−0.10
Brittleness	0.31 *	−0.31 *	−0.02	0.06	−0.07	0.37 *	−0.11
Adhesiveness	0.40 *	−0.23	−0.14	0.18	0.01	0.27	−0.12
E	−0.40 *	0.22	0.15	−0.18	−0.01	−0.26	0.12
Consistency	−0.29 *	−0.31 *	0.00	0.04	−0.08	0.37 *	−0.10

SR—serum release; WL—water content measured as weight loss (WL) during drying; E—Young’s modulus. ^1^ Sensory attributes: 1—hardness; 2—brittleness; 3—moisture; 4—springiness; 5—adhesiveness; 6—chewiness; 7—easy to swallow. * *p* < 0.05.

**Table 7 gels-09-00225-t007:** Pearson’s correlations among liking and sensory attributes of PGs.

	Liking and Sensory Attributes										
	1	2	3	4	5	6	7	8	9	10	11
Overall liking	-										
Consistency liking	0.65 *	-									
Aroma	0.03	0.04	-								
Taste	0.36 *	0.22	0.35 *	-							
Hardness	−0.07	−0.22	0.08	0.08	-						
Brittleness	0.22	0.39 *	0.41 *	0.50 *	−0.12	-					
Moisture	0.05	0.37 *	−0.14	0.17	−0.32 *	0.31 *	-				
Springiness	−0.16	0.12	0.43 *	0.07	0.27	0.23	0.19	-			
Adhesiveness	0.14	−0.03	0.10	0.17	−0.12	0.18	0.18	−0.01	-		
Chewiness	−0.13	−0.20	−0.06	−0.06	0.62 *	−0.20	−0.27	0.05	−0.10	-	
Ease to swallow	0.32 *	0.43 *	0.18	0.24	−0.42 *	0.28	0.33 *	−0.01	0.04	−0.36 *	-

* *p* < 0.05.

**Table 8 gels-09-00225-t008:** Pearson’s correlations between sensory attributes and oral-processing parameters of PGs.

	Sensory Attributes ^1^						
	1	2	3	4	5	6	7
Duration	−0.02	−0.17	−0.28	−0.02	−0.05	0.07	−0.12
Number of chews	−0.02	−0.11	−0.27	−0.05	0.05	0.12	−0.13
MA (mV)	0.23	−0.04	−0.17	0.27	−0.05	0.31 *	−0.46 *
TA (mV)	0.23	−0.13	−0.19	0.29 *	−0.13	0.17	−0.55 *
MA (mV × s)	0.12	−0.10	−0.23	0.12	−0.06	0.23	−0.38 *
TA (mV × s)	0.12	−0.17	−0.28	0.13	−0.13	0.20	−0.42 *
Saliva uptake	0.23	−0.07	−0.16	0.03	0.02	0.36 *	0.08
Saliva incorporation rate	0.22	−0.02	−0.02	0.02	0.01	0.26	−0.04
Viscosity of bolus:							
ƞ10	−0.19	0.01	0.26	−0.06	0.14	−0.30 *	0.12
ƞ55	−0.10	−0.01	0.14	−0.10	0.34 *	−0.20	−0.06

MA and TA—masseter and temporalis muscle activity, respectively. ^1^ Sensory attributes: 1—hardness; 2—brittleness; 3—moisture; 4—springiness; 5—adhesiveness; 6—chewiness; 7—easy to swallow. * *p* < 0.05.

**Table 9 gels-09-00225-t009:** Chemical characteristics of AP and HP.

Pectin	Monosaccharides (mol%) ^a^						Rha/GalA	RG-I% ^b^	(Ara + Gal)/Rha ^c^	DM ^d^	M_w_, kDa	M_w_/M_n_
	UA	Gal	Xyl	Glc	Rha	Ara						
HP	90.6 ± 0.7	3.2 ± 0.2	1.0 ± 0.3	0.5 ± 0.2	2.1 ± 0.1	3.3 ± 0.1	0.02	10.61	3.13	21	538	4.1
AP	89.5 ± 0.7	2.6 ± 2.6	3.8 ± 0.1	1.7 ± 0.1	1.6 ± 0.1	0.8 ± 0.5	0.02	6.61	2.14	43	401	5.2

^a^ Data are calculated as molar percentages. ^b^ Rhamnogalacturonan I = 2Rhamnose% + Arabinose% + Galactose%. ^c^ Average length of rhamnogalacturonan I side chains. ^d^ Degree of methyl esterification. UA: uronic acids; GalA: galacturonic acid; Gal: galactose; Xyl: xylose; Glc: glucose; Rha: rhamnose; Ara: arabinose.

**Table 10 gels-09-00225-t010:** Final compositions of the five pectin gel samples.

Hydrogel	AP (*w*/*v* %)	HP (*w*/*v* %)	Sucrose (*w*/*v* %)
PG1	4	-	10
PG2	3	1	10
PG3	2	2	10
PG4	1	3	10
PG5	-	4	10

## Data Availability

The data that support the findings of this study are available from the corresponding author upon reasonable request.

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
