# Peer review of "Effect of Hogweed Pectin on Rheological, Mechanical, and Sensory Properties of Apple Pectin Hydrogel"

_gels, 2023, doi:10.3390/gels9030225_

Round 1

Reviewer 1 Report

Food gels prepared from two  low-methyl-esterified pectin, apple pectin (AP) and hogweed pectin (HP) and their mixtures were prepared and characterized, demonstrating the hogweed pectin feasibility as food ingredient. Some methods are not clearly presented. Some results sections are too long and can be divided into more sections. Moreover, there are some issues which should be corrected such as

 Regarding The abstract

- should be a total of about 200 words maximum. 

In line 13 you can remove word five.

For The synergistic gelation between pectins occurred because Young’s modulus and tangent after flow point values were higher for mixed hydrogels than for pure AP and HP hydrogels.

I suggest a reformulation in order to reduce the number of words: Young’s modulus and tangent after flow point values were higher for mixed hydrogels than for pure AP and HP hydrogels suggesting a synergistic effect.

The synergistic effect was revealed due to this analysis, it do not occurred due to them.

Please find another term for liking evaluation.

The methods employed in the study are not clearly stated in the abstract. They can be deducted from the results. Please rewrite the abstract.

If LM pectin stands for low-methyl-esterified pectins, please introduce the abbreviation in the abstract.

 Regarding The introduction

In lines 50, 58-60 please introduce a reference.

Results
Hydrogel production is not clear in lines 95. Please mention the water amount.

In figure 1 please mention which sample is illustrated.

Information in lines 104-106 should be moved to material methods section, along with the figure 1.

Please extend the discussion of the results regarding the ph, desnity and water content.

Lines 126-129 Small deformation dynamic rheological measurements were conducted by applying  a small oscillating stress or strain and recording the responses of the hydrogels. Strain  sweep experiments were performed from 0.01 to 100% strain amplitude using a controlled shear rate mode at 20°C at a constant frequency and stress of 1 Hz and 9.0 Pa,  respectively.

Should be moved to methods section.

In figure 2 the differences stated for the GLVR are not visible. Maybe all the results should be overlapped, as in the figure representing the puncture test. However, the information in table is already enough so the figure is not necessary in my opinion.

Line 213 please add reference.

Correlation analysis can be presented in a different section.

Materials and methods

In lines 480 please introduce the notation used in lines 104.

The results regarding the weight loss of the samples are not clear. Are they expressed as water content (80.7-82.7%)?

The density determination is not presented.

References

Lines 677 check Gels name.

682 bold year

691 Molecules

693 bold year

Lines 702, lines 773,776,788 bold year

Lines 793 change font

However, I recommend the acceptance with minor revision due to the consistency of the paper.

Reviewer 2 Report

The current manuscript entitled "Effect of hogweed pectin on rheological, textural, and sensory  properties of apple pectin hydrogel" characterizes a food gel prepared from two low-methyl-esterified pectins, apple pectin  and hogweed pectin . It is a well-written manuscript and can be considered for the publication after minor correction. My specific comments are attached as a PDF file. Thank you. 

Reviewer 3 Report

Comments in attachment file
